# Association between Moving to a High-Volume Hospital in the Capital Area and the Mortality among Patients with Cancer: A Large Population-Based Cohort Study

**DOI:** 10.3390/ijerph18073812

**Published:** 2021-04-06

**Authors:** Jung-kyu Choi, Se-Hyung Kim, Myung-Bae Park

**Affiliations:** 1Institute of Health Insurance & Clinical Research, National Health Insurance Service Ilsan Hospital, Goyang 10444, Korea; yolong21@nhimc.or.kr; 2National Medical Center, National Emergency Medical Center, Seoul, Seoul 04564, Korea; kurukuru5@nmc.or.kr; 3Department of Gerontology Health and Welfare, Pai Chai University, Daejeon 35345, Korea

**Keywords:** patient movement, mortality, cancer, income

## Abstract

This study aimed to identify the association between moving to a high-volume hospital and the mortality of patients with cancer living in the district. The study population comprised participants diagnosed with cancer within the past nine years (2004–2012). The final sample included 8197 patients with cancer, 3939 were males (48.1%), and 4258 were females (51.9%). A Cox proportional hazard model was used to estimate the hazard ratio (HR) for death. Confounding variables including sex, age, type of social security, income level, disability, and utilization volume were incorporated into the model. Among patients with cancer living in the district, 2874 (35.1%) used healthcare services in Seoul. About 10% (*n* = 834) of patients died during the follow-up period. The HR for death in females (HR: 0.68, 95% CI: 0.58–0.81) was lower than that in males. Additionally, the HR for the death of patients using healthcare services in Seoul (HR: 1.30, 95% CI: 1.11–1.53) was higher than those patients who did not use healthcare services in Seoul. Among patients utilizing services in the province, wealthier patients’ survival probability was significantly higher than that of others. The cause of income differences should be identified, and accessibility to medical use of low-income families should be enhanced to prevent mortality of patients from cancer disparities.

## 1. Introduction

The incidence and mortality of cancer are expected to increase rapidly because of the aging population. In Korea, more than 217,000 patients were newly diagnosed with cancer in 2014, and more than 76,000 died of cancer. The number of deaths from cancer account for about 29% of all deaths [1]. The economic burden of cancer increased with an annual growth rate of 8.9% during 2000–2010 [2]. The economic burden of treating cancer has continued to grow over time. This trend is thought to result from an increase in life expectancy and the overall aging of the Korean population. This is a burden not only for individuals but also for the nation as a whole.

Patients must obtain a referral, easily issued from a local hospital through the healthcare delivery system, for them to receive treatment services at high-volume hospitals in the capital area or elsewhere. Patients are permitted to go to any doctor or medical institution within Korea to receive treatment without restraining healthcare services [3,4]. Indeed, due to recent improvements in transportation, many cancer patients who lived in district areas have moved to the capital and use medical care [5]. Patients with severe diseases may migrate from districts to the metropolitan area to use the high-volume hospitals in those regions [6,7,8]. In other words, it means that many patients with cancer living in a district area are being treated at high-volume hospitals in Seoul, the capital of Korea. Thousands of preventable deaths occur because high-risk surgery is performed in hospitals with little experience with surgical procedures. According to several studies, high-volume hospitals exhibit lower mortality rates than low-volume hospitals for specific conditions [9] because most tertiary hospitals have high-tech equipment and can train top-level practitioners; however, most of them are located in cities. Especially in Korea, all of the famous tertiary hospitals are concentrated in Seoul [10]. When patients are diagnosed with severe disease and intend to go to the capital to use advanced healthcare technology and skills, they must travel to obtain these services and satisfy their care needs. In general, cancer patients are likely to die within several years. Therefore, many of them prefer famous and high-volume hospitals in Seoul.

Disparities in cancer care exist throughout, from cancer prevention and early detection to treatment, survivorship, and palliative care. Disparities in income level, health insurance status, and race or ethnicity are risk factors for an increased chance of death in patients with cancer. Numerous studies identified large disparities in cancer burden according to socioeconomic status. Socioeconomic factors such as income level, geographical residence, and inadequate education were found to be far more important than biological differences [11,12,13,14,15,16,17,18]. It is necessary to eliminate improvable disparities to reduce cancer incidence and mortality and to increase the cancer survival rate of the socially and economically disadvantaged to a similar level to that of the general population. Several studies examined the effects of moving to high-volume hospitals on the death of patients [19,20]. Studies show conflicting findings. The lack of access to healthcare services might adversely affect cancer incidence and mortality throughout the continuum from cancer prevention and early detection to treatment, survivorship, and palliative care. Moreover, the burden on family caregivers as well as the quality of life of cancer patients due to long-distance travel for treatment is increasing. This is becoming a new aspect of inequality for cancer patients living in district areas. Differences in the use of medical services and the mortality of patients with cancer are of considerable importance to policy. Thus, it is necessary to establish the survival rate of cancer patients who travel long distances for treatment, but few studies are related to this. Our study aims to identify patients with cancer living in the region using high-volume hospitals in the capital. The study likewise seeks to confirm whether such medical use affects their mortality through nationwide population-based cohort data. Furthermore, our analysis identified the status of medical use and the associated income level and death.

## 2. Methods

### 2.1. Data and Participants

The National Health Insurance Service (NHIS) established a nationwide cohort containing medical care claims from 2002, the baseline year. The cohort population includes 1,025,340 people, comprising about 2% of the total Korean population. The cohort was followed until 2013. A stratified sampling method was used to divide subjects by sex, age, and income level. Sex and age were grouped into two (male, female) and 18 (0, 1–79 (5-year increments), 80+) categories. Income level was grouped into 41 categories (medical aid: 1, industrial worker (IW): 20, self-employed (SE): 20). A total of 1476 stratified categories were present.

The study population comprised patients living more than 200 km from the capital and those newly diagnosed with cancer (C code) according to the International Classification of Diseases 10th revision (ICD-10) during a nine-year period (2004–2012). Patients (*n* = 2899) treated for cancer in 2002 and 2003 were excluded in the analysis. Patients with cancer (*n* = 1303) newly diagnosed in 2013 were likewise excluded from the analyses. Patients (*n* = 1255) who died within one year of the cancer diagnosis were also excluded from the analyses. This study is designed for patients who were observed for at least one year. No information exists on the cancer stage in the data. The cancer stage is a decisive factor that influences death. Patients with more advanced cancer stages are more likely to seek care in the capital’s major hospitals. As such, patients who died within one year of diagnosis were excluded. Thus, the final sample included 8197 patients with cancer (Figure 1): 3939 males (48.1%) and 4258 females (51.9%). The average follow-up period was 4.7 years (range: 1–10, median: 4.1).

### 2.2. Variables

One of the outcome variables was death. Variables considered to be confounders in the multivariate model were sex, age, income level, type of social security, utilization volume, chemotherapy, and disability. Age was divided into five groups (≤39, 40–49, 50–59, 60–69, ≥70). The type of social security consisted of medical aid and health insurance (industrial worker, self-employed). According to insurance premiums, income level was divided into five quintiles (1: lowest, 5: highest). The type of social security was divided into three groups. All people in South Korea are eligible for coverage under the National Health Insurance Program. The insured are divided into two groups: industrial worker and self-employed. Besides, Medical Aid exists, which provides almost free medical support in the country. Utilization volume consisted of outpatient visits, inpatient visits, and length of stay (LOS). In our study, we defined the utilization volume only for medical use at tertiary hospitals in Seoul. In Korea, to be designated as a tertiary referral hospital, the hospital’s size must be considerable and tertiary care must be implemented, and all these standards are stipulated by medical law. The Korean government selects 13 tertiary hospitals in Seoul every three years. The variables associated with the utilization volume mentioned above are continuous. Relevant chemotherapeutic drugs were prescribed after the cancer diagnosis. The disability category was divided into groups with or without disability. A major independent variable is the dummy variable indicating the use or non-use of healthcare services in the capital. This is derived from confirming all medical use after cancer diagnosis through claim data. This variable demonstrated only two possible values: 0 = did not use services in the capital, and 1 = used services in the capital. Claim data includes the location and type of hospital. We divided into two groups—one that used medical care at a tertiary hospital in the capital city for cancer treatment and management at least once after a cancer diagnosis, and one that never used medical care in Seoul.

### 2.3. Statistical Analysis

Descriptive statistics were used to evaluate differences in demographic characteristics. The chi-square test was used to compare the survival rate depending on the use or not of the capital’s healthcare services. A Cox proportional hazard model was used to estimate the hazard ratio (HR) for death. Covariates were sex (ref: male), age (ref: under 40), social security type (ref: medical aid), income level (ref: 1 quintile), disability (ref: normality), and utilization volume (continuous). The survival rate was calculated using a Kaplan–Meier curve of the follow-up period. The SAS statistical package version 9.4 was used to perform the analysis in this study. A *p*-value < 0.05 was considered to be significant.

## 3. Results

The characteristics of patients with cancer are shown according to whether or not they use healthcare services in the capital (Table 1). Among the population living in provinces, 8197 patients were newly diagnosed with cancer. A total of 2874 patients (35.1%) were using healthcare services in the capital. About 10% (*n* = 834) of the patients with cancer died during the follow-up period. The survival and death rates among patients using services in the capital were 34.4% and 40.8%, respectively.

Males (37.9%) using services in the capital outnumbered females (32.4%). The proportion of patients with cancer using services in the capital peaked at 50 to 59 years (38.9%). However, the proportion decreased among those aged 70+ years (27.8%). A total of 25.2% of patients were in the first quintile of household income, and 43.1% were in the fifth quintiles. Such indicates an increasing number of patients with increasing income using healthcare services in the capital. The rate of patients using healthcare services in the capital was higher in industrial workers (36.1%) and the self-employed (36.1%) categories than in the medical aid category (16.7%). The rate of patients without disability (35.5%) using healthcare services in the capital was higher than that of patients with disability (31.3%). The proportion of patients who received chemotherapy and those who died were high in the group that was treated in the capital. Significant differences were found in sex, age, income level, type of social security, disability, chemotherapy, and death.

Table 2 shows the number of fatalities in patients with cancer according to the use of services in Seoul by the five most prevalent cancers. The prevalence rate was highest for digestive organs (40.3%), followed by the thyroid and other endocrine glands (18.2%), breast (8.5%), respiratory and intrathoracic organs (8.2%), and female genital organs (5.5%). Death is strongly correlated with the use of services in Seoul for cancer of the thyroid and other endocrine glands and female genital organs (*p* < 0.05).

Table 3 compares the utilization (outpatient visit, inpatient visit, and LOS) volume of healthcare services between patients using healthcare services in the capital and patients who did not. The number of outpatient visits for patients (17.53 times) using healthcare services in the capital is significantly higher than that of patients (15.09 times) who did not. Moreover, the number of inpatient visits for patients (5.19 times) using healthcare services in the capital is significantly higher than that of patients (3.78 times) who did not. LOS was similar between the two groups. Patients using healthcare services in the capital used most (80%) of their cancer-related medical use in the capital.

Table 4 shows the factors associated with the mortality of patients with cancer. In a univariate model, significant variables associated with mortality were sex, age, disability, utilization volume, and use of services in the capital. Conversely, in a multivariate model, significant variables associated with mortality were sex, age, and utilization of services in the capital. The HR for death in females (HR: 0.68, 95% CI: 0.58–0.81) was lower than that in males. The HRs for death steadily increased by age group when compared to the reference group (under 40 years of age). The HR for death in patients with disability (HR: 1.27, 95% CI: 1.03–1.57) was higher than that of the reference group (without disability). The HRs for death increased by utilization volume related with the inpatient category (visit and LOS). The HR for death of patients using services in the capital (HR: 1.30, 95% CI: 1.11–1.53) was higher than in patients who did not use services in the capital.

Figure 2 shows the results of the Kaplan–Meier survival curves, illustrating mortality after a cancer diagnosis. Income level was divided into two categories: 1–4 quintiles and 5 quintiles. Among patients utilizing services in the province, the survival probability of richer patients (5 quintiles) was significantly higher than that of other patients (1–4 quintiles). However, income level did not affect death among patients utilizing services in the capital.

## 4. Discussion

Several studies have been conducted to identify cancer disparities throughout the continuum. Such studies cover cancer prevention and early detection to treatment, survivorship, and palliative care [11,12,13,14,15,16,17,18]. In practice, the removal of cancer disparities is defined as reducing cancer incidence and mortality and increasing cancer survival among vulnerable social groups [21]. In the past decade, transportation facilities have been developed to increase the use of healthcare services in the capital brought about by industrialization and economic development [21,22]. This facilitated the movement of patients with cancer to the capital [5]. However, few studies have examined the status of medical use used in the capital by patients with cancer living as residents in district areas. This article also presents data on the association between hospital movement and survival for cancer-based analyses of nationwide population-based cohort data. Differences in cancer mortality are present, depending on the use of services in the capital. According to our results, the risk of dying was greater for patients with cancer treated in the capital than for patients with cancer not treated in the capital.

The proportion of patients who used healthcare services in the capital was 35.1% among patients with cancer who lived in the province. Among these patients, the prevalence rate was highest for cancer of the digestive organs. Death is strongly correlated with the use of services in Seoul for cancers of the thyroid and other endocrine glands and female genital organs. Cancers of the thyroid and other endocrine glands and female genital organs exhibit a higher survival rate than other cancers. Thus, they demonstrate a good prognosis in Korea [1]. However, patients with these cancers who were treated in the capital exhibit poor survival rates because they may be in a serious condition.

Approximately 10% of patients with cancer died during the follow-up period of 4.7 years. The number of outpatient and inpatient visits for patients with cancer treated in the capital was significantly higher than those treated in the provinces. Patients who used healthcare services in the capital constituted most of the utilization volume in the capital.

In a multivariate model, risk factors for death are sex, age, disability, utilization volume, and use of services in the capital. The risk of dying was 0.69 times (95% CI: 0.58–0.81) lower for females than for males. Recent studies noted higher cancer incidence and mortality among males compared to females [23,24,25]. According to our results, the risk of death in patients with cancer gradually increased with age. Several studies demonstrated disparities in cancer survival by insurance status [11,14]. Cancer mortality started to decrease as medical technology advances. However, cutting edge medical technology is expensive because procedures and drugs are not covered by the NHIS. Wealthier patients can afford expensive and powerful procedures and drugs that are not covered. This is the reason why they come to the capital for treatment [22,26]. Poorer survival appears to result more from the disparities in access to care and the quality of cancer treatment [27]. Studies on treatment outcome in settings where all patients with cancer receive equal access to treatment and supportive care confirmed that similar treatments yield similar outcomes [28,29]. However, in this large national study, income, as well as type of social security, did not affect patients with cancer death. Such may be because Korea is a country with universal health coverage and exhibits a benefits system that provides adequate treatment for cancer [30]. The range of services provided by the country is the same for medical aid patients. The risk of dying was 1.30 times (95% CI: 1.11–1.53) greater for patients using services in the capital than for patients using services in the province. Patients with cancer treated in the capital might be in a more serious condition than those treated in the provinces. Patients with severe cancer who live in the provinces may prefer to use high-quality healthcare services in the capital. Researchers reported that 26% of patients with cancer living in the provinces used the five largest hospitals in Seoul. This means that Korea exhibits a very high preference for the five largest hospitals. Moreover, it is likely that they used these hospitals because of the severity of their cancer. In Korea, sex and age demonstrate a major impact on cancer mortality. Our findings reinforce the existing research that the biological or behavioral characteristics of sex are more important factors than socioeconomic factors such as region, income, and type of social security.

The survival probability of the rich was significantly higher than the survival probability of other patients utilizing services in provinces. However, no association exists between survival probability and income level among patients with cancer utilizing services in Seoul. Lower survival appears to result in more from disparities in access to care and quality of cancer treatment [27]. Besides, the transportation cost burden on people living in rural areas and low-income populations are higher. Therefore, patients in these vulnerable groups carry a double burden from cancer [31]. Studies on treatment outcomes in settings where all patients with cancer receive equal access to treatment and supportive care confirmed that similar treatments yield similar outcomes [28,29]. Income is a factor that affects the use of hospitalization and outpatient services, the amount of medical expenditure, and the type of medical facility [32].

This study exhibits several limitations because it relied on data from claims. No information exists on cancer severity or cancer sites. Cancer severity is a decisive factor that influences mortality and the use of healthcare services. Patterns of use might differ by the severity of cancer. Patients with severe cancer who died within one year of cancer diagnosis were excluded from the analyses to overcome the limitation. The utilization volume and chemotherapy sessions were included in the multivariate model to adjust cancer severity. Second, other medical conditions can also affect the results. Therefore, considering this variable can lead to more powerful evidence [33]. Third, we did not attempt to adjust for characteristics, such as provider skills, which are likely to be highly correlated with mortality. High-volume providers achieve better outcomes than low-volume providers [34,35]. In conclusion, to clarify the causality, the homogeneity of the experimental group and the control group of the model should be secured and compared. Despite these limitations, this study demonstrated an association between hospital movement and mortality in patients with cancer using representative population-based follow-up data.

## 5. Conclusions

The centralization of cancer treatment can significantly increase the patients’ travel distance, especially those living in rural areas. It might lead to treatment delays for many people. Patients at high-volume hospitals waited longer for treatment than those in low- to medium-volume hospitals [36]. Consequently, it can result in a negative effect on the patient’s prognosis. High-quality healthcare services should be distributed equally across districts. The mortality rate differed significantly by income level, which provides essential implications to policymakers. Addressing low access to healthcare services in vulnerable social groups such as low-income groups is not the only solution that is needed to remove cancer treatment disparities. Policymakers should increase the coverage rate for cancer services that are currently not covered. Further study should identify any risk factors related to death in patients with cancer to eliminate treatment disparities.

## Figures and Tables

**Figure 1 ijerph-18-03812-f001:**
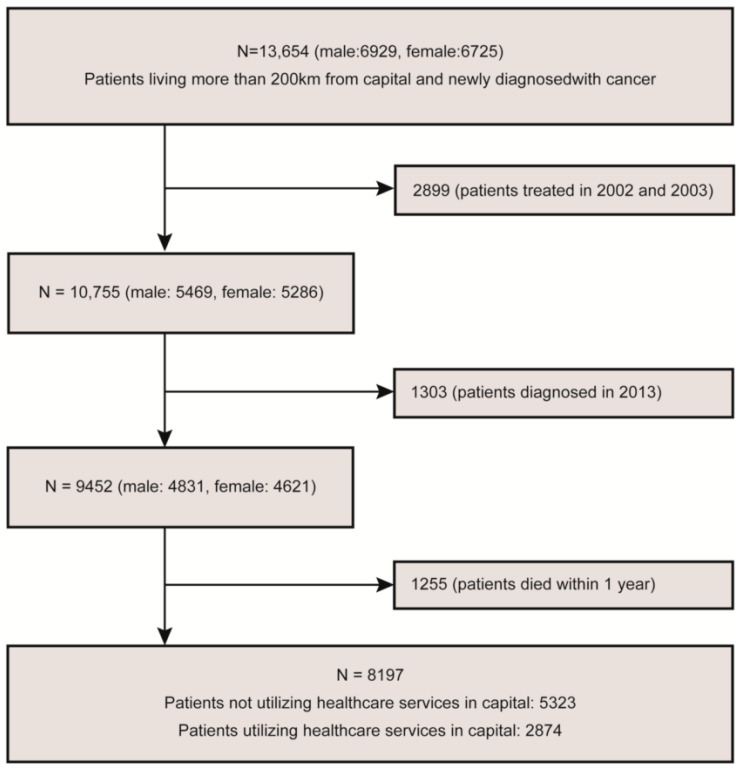
Flow chart of the patients included in the analysis.

**Figure 2 ijerph-18-03812-f002:**
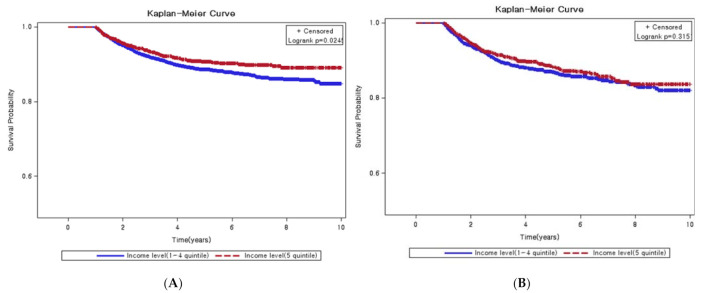
Kaplan–Meier curve of cancer patients for ten years according to income level by utilization in the capital. (**A**) Patients not utilized in the capital; (**B**) Patients utilized in the capital.

**Table 1 ijerph-18-03812-t001:** Characteristics of the study population.

		Total	Patients Not Utilizing Healthcare Services in Capital		Patients Utilizing Healthcare Services in Capital		*p*-Value
			n	%	n	%	
Total		8197	5323	(64.9)	2874	(35.1)	
Sex	Male	3939	2445	(62.1)	1494	(37.9)	<0.0001
	Female	4258	2878	(67.6)	1380	(32.4)	
Age Group (Years)	≤39	908	576	(63.4)	332	(36.6)	<0.0001
	40‒49	1379	882	(64.0)	497	(36.0)	
	50‒59	2099	1282	(61.1)	817	(38.9)	
	60‒69	2075	1329	(64.0)	746	(36.0)	
	≥70	1736	1254	(72.2)	482	(27.8)	
Income Level	1 grade (lowest)	1413	1057	(74.8)	356	(25.2)	<0.0001
	2 grade	1045	738	(70.6)	307	(29.4)	
	3 grade	1392	930	(66.8)	462	(33.2)	
	4 grade	1754	1123	(64.0)	631	(36.0)	
	5 grade (highest)	2593	1475	(56.9)	1118	(43.1)	
Type of Social Security	Medical aids	442	368	(83.3)	74	(16.7)	<0.0001
	Industrial worker	4969	3174	(63.9)	1795	(36.1)	
	Self-employed	2786	1781	(63.9)	1005	(36.1)	
Disability	Not disabled	7318	4719	(64.5)	2599	(35.5)	0.013
	Disabled	879	604	(68.7)	275	(31.3)	
Chemotherapy	No	916	686	(74.9)	230	(25.1)	<0.0001
	Yes	7281	4637	(63.7)	2644	(36.3)	
Death	No	7363	4829	(65.6)	2534	(34.4)	< 0.0001
	Yes	834	494	(59.2)	340	(40.8)	

Data are expressed as N (%).

**Table 2 ijerph-18-03812-t002:** Survival and death by cancer according to the use of health care in Seoul.

	Total	Survival		Death		*p*-Value
		n	%	n	%	
Digestive Organs	3304	2856	(86.4)	448	(13.6)	
Not Utilizing in Capital	1977	1719	(86.9)	258	(13.1)	0.2967
Utilizing in Capital	1327	1137	(85.7)	190	(14.3)	
Thyroid and Other Endocrine Gland	1489	1483	(99.6)	6	(0.4)	
Not Utilizing in Capital	1106	1104	(99.8)	2	(0.2)	0.0215
Utilizing in Capital	383	379	(99.0)	4	(1.0)	
Breast	694	656	(94.5)	38	(5.5)	
Not Utilizing in Capital	471	448	(95.1)	23	(4.9)	0.3189
Utilizing in Capital	223	208	(93.3)	15	(6.7)	
Respiratory and Intrathoracic Organs	670	534	(79.7)	136	(20.3)	
Not Utilizing in Capital	429	345	(80.4)	84	(19.6)	0.5375
Utilizing in Capital	241	189	(78.4)	52	(21.6)	
Female Genital Organs	454	428	(94.3)	26	(5.7)	
Not Utilizing in Capital	314	303	(96.5)	11	(3.5)	0.0023
Utilizing in Capital	140	125	(89.3)	15	(10.7)	

Data are expressed as N (%).

**Table 3 ijerph-18-03812-t003:** Utilization volume of patients with cancer.

	Patients not Utilized in Capital		Patients Utilized in Capital		*p*-Value
	Mean	SD	Mean	SD	
Outpatient Visit (no.)	15.09	14.61	17.53 (14.02)	16.90	<0.0001
Inpatient Visit (no.)	3.78	5.35	5.19 (4.37)	7.94	<0.0001
Length of Stay (days)	30.88	44.99	29.32 (22.17)	38.65	0.2980

SD = standard deviation. The contents in parentheses are the amount of use in the capital.

**Table 4 ijerph-18-03812-t004:** Factors related to survival of cancer patients by a Cox proportional hazard model.

		Univariate Model				Multivariate Model			
		HR		95% CI		HR		95% CI	
Sex (ref: Male)	Female	0.42 ***	0.36		−0.48	0.69 ***	0.58		−0.81
Age Group (ref: ≤39)	40–49	1.62 *	1.07		−2.44	1.67 *	1.07		−2.61
	50–59	2.59 ***	1.78		−3.78	2.09 ***	1.39		−3.14
	60–69	4.01 ***	2.78		−5.78	3.16 ***	2.12		−4.72
	≥70	4.93 ***	3.42		−7.11	4.33 ***	2.87		−6.55
Income level (ref: 1 Quintile)	2 quintile	0.96	0.74		−1.24	0.89	0.66		−1.21
	3 quintile	1.13	0.90		−1.43	1.06	0.81		−1.39
	4 quintile	1.20	0.97		−1.49	1.14	0.88		−1.47
	5 quintile	0.93	0.75		−1.15	0.89	0.69		−1.14
Type of Social Security (ref: Medical Aid)	IW	1.32	0.92		−1.91	1.13	0.72		−1.78
	SE	1.37	0.94		−1.99	1.13	0.71		−1.78
Disability (ref: Normal)	Handicapped	1.52 ***	1.25		−1.84	1.27 *	1.02		−1.57
Outpatient Visit(no.)	continuous	1.02 ***	1.02		−1.02	1.00	1.00		−1.00
Inpatient visit(no.)	continuous	1.03 ***	1.03		−1.04	1.02 ***	1.01		−1.03
Length of Stay(days)	continuous	1.01 ***	1.01		−1.01	1.01 ***	1.00		−1.01
Chemotherapy (ref: No)	Yes	18.60 ***	7.73		−44.77	16.07 **	2.26		−114.34
Utilized in Capital (ref: No)	Yes	1.23 ***	1.07		−1.42	1.29 **	1.10		−1.51

HR = hazard ratio, CI = confidential interval, IW = industrial worker, SE = self-employed. The *p*-value is for testing of variables that may affect the death. It is desirable to interpret the significance level of statistical decision-making down to 2.5% by Bonferroni’s correction to control family-wise type 1 error rate. *** *p* < 0.001, ** *p* < 0.01, * *p* < 0.025.

## Data Availability

The data is from the National Sample Cohort (data No.: 2018-2-200) provided by the NHIS.

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
