# Peer review of "Association between Moving to a High-Volume Hospital in the Capital Area and the Mortality among Patients with Cancer: A Large Population-Based Cohort Study"

_ijerph, 2021, doi:10.3390/ijerph18073812_

Round 1

Reviewer 1 Report

First of all, congratulate the authors for choosing such an interesting and important topic for health as cancer.

However, I would like to make a few comments to help improve the quality of this manuscript.

The first aspect that strikes me is the title of the manuscript. "Impact of the sex difference ...". As explained between lines 53 and 55, the purpose of the study is to identify the association between medical use and death in cancer patients. Therefore, sex is not the most important aspect of the study and I think it should not appear in the title. I would suggest modifying the title.

Second, I ran into an error dividing the age into groups. The first group is described as <39. I think it should actually be written as <40. If expressed as <39, people between 39 and 40 would not belong to any group. See line 86, table 1 and table 4.

Third, I think the results were not stated clearly enough. I would suggest developing and better explaining the multivariate model.
I understand that mortality is the dependent variable and that sex, age, income level, and use of health services in the capital are the independent variables. Therefore, I would like to see the different correlations between the dependent variable with the independent ones. I even think it would be very interesting to build a binary logistic regression model to establish associations between the different independent variables and the dependent variable.

In the discussion section I have detected several statements that lack their corresponding bibliographic citations. For example:
- Line 164-166 "Several ..... care."
- Line 169-172 "In the past .... services"

Conclusions section. Conclusions should be based on the results obtained. I think this section concludes with a series of statements that do not have much to do with the results obtained. I would advise that the conclusions be rewritten based on the results obtained.

Kind regards

Author Response

Thank you for your valuable comments. Please see the attached file (our response to reviewer).

Reviewer 2 Report

The theme is interesting, but the findings are too weak.

The main findings are affected by possible bias, like severity of cancer that influences survival analysis and the patient choices to ask health cure in Seoul.

In other words, I'd suggest to include in the analysis other confounding factors to avoid spurious correlations

Author Response

Thank you for your valuable comments. Please see attached file (our response).

Reviewer 3 Report

Thank you for the chance to review this interesting manuscript. Generally, the manuscript is well-written and offers an unique viewpoint in utilization of the healthcare.

Considering the emphasis in the title and conclusions on sex differences, there was surprisingly little literature included about the any previous research regarding the sex differences in utilization of cancer healthcare services. Authors should consider adding some literature and include this also in discussion. Additional point, the authors refer in a number of occasions to individuals’ beliefs that treatment in capitol Seoul is superior to that in provincial towns and cities. Please could you add a reference. Further minor comments included below.

Abstract

Line 25: “…cancer disparities…“ – unclear what this term refers to, please clarify

Introduction

Line 39: “…migrate…” – please clarify – does this mean in a sense of moving physically to another district or rather deciding to use services in another district while continuing to live in the place of residence prior to diagnosis.

Line 40: “Thousands of preventable deaths occur because high-risk surgery is performed in hospitals that have little experience with surgical procedures.” – Please could you clarify in the text whether this is specific in Korea or internationally.

Line 47: “…insurance status…” – please could you add “health” insurance status – presumably this refers to the coverage of the health insurance.

I am slightly struggling with the term of “transfer” in this context. Based on the information in the manuscript, after the cancer diagnosis (primary care doctor??), patients have a free choice of where they decide to get treated – before the treatment has commenced. Therefore, this is really not a care transfer, rather a decision of an individual to be treated elsewhere than in their nearest available (appropriate) hospital/care centre.

Line 53: “The purpose of this study was to identify the association between medical use and the death of patients with cancer in the district area using nationwide population-based cohort data in Korea.” – Currently this aim is very difficult to understand – what is meant with medical use (of what? Different hospitals?)

Methods

Line 55: “In addition, our analysis identified the status of medical use and the effect of income level” – Please rephrase- this reads as a result not as an aim.

Line 59: Please use the word participants throughout instead of subjects (subject refers to someone subject to a monarch such as king or queen).

Also – were the participants excluded if they moved within 200km of the capital during the follow-up period?

Line 87: “The type of social security consisted of medical aid and health insurance (industrial worker, self-employed)”. Please could the authors add few words of explanation what the different types of insurances mean for readers not familiar with Korean’s social security system (e.g. does self-employed mean private insurance? Medical aids – publicly funded?).

Considering that many of the variables were used in more than one analysis, did authors consider adjusting significance level? Also, was any ethical approval sought/needed for the study?

Results

Line 117: “…in the capital increased to 60 years…” – please could you rephrase, this is difficult to understand

Line 147: “Patients using healthcare services in Seoul comprise most of the utilization volume in Seoul.” – Sorry, I don’t quite understand this sentence.

Line 194: “Wealthier patients…” – While undisputed, please add reference. While income was included in the model, use of extra services was not evaluated in the current research.

Discussion

Line 202: Please add the reference

Line 216: Please add the reference

Author Response

Thank you for your valuable comments. Please see attached file(our response).

Round 2

Reviewer 1 Report

Great improvements can be seen in this second version of the manuscript.

The authors have followed most of the recommendations provided by the reviewers and I believe that this version would be acceptable for publication.

Thank you very much

Regards

Author Response

We really thank you for your thoughtful suggestions and insights. 

Reviewer 2 Report

This paper have yet some weakness depending by few data available.

Maby there is a mistake in 173 line "In a multivariate model, significant variables influencing mortality were sex, age, income level, and utilization of services in the capital". Infact, income level is not significant on risk factors.

Author Response

We are sorry about our mistake. Currently, we revised a wrong sentence, and we excluded income as you point out.

The manuscript has benefited from your insightful suggestions.